# Vision Therapy: A Primer and Caution for Pediatricians

**DOI:** 10.3390/children9121873

**Published:** 2022-11-30

**Authors:** Bo Wang, Edward Kuwera

**Affiliations:** 1Department of Ophthalmology, Duke University Medical Center, Durham, NC 27710, USA; 2The Wilmer Eye Institute, Department of Pediatric Ophthalmology and Adult Strabismus, The Johns Hopkins University School of Medicine, Baltimore, MD 21287, USA

**Keywords:** vision therapy, behavioral optometry, visual therapy, behavioral therapy, syntonics, binasal occlusion, convergence insufficiency, anti-suppression

## Abstract

Vision therapy, also known as behavioral therapy, is theorized by its practitioners to treat a variety of visual disorders, including learning disability in children. However, the utility of vision therapy to treat various learning disabilities is challenged by the American Academy of Pediatrics, American Academy of Ophthalmology, American Association for Pediatric Ophthalmology and Strabismus, and the American Association of Certified Orthoptists. The purpose of this review is to (1) provide an overview of vision therapy, (2) evaluate the evidence for vision therapy, and (3) give practical recommendations for pediatric primary care providers regarding vision therapy. A review of the literature demonstrates evidence that vision therapy is useful in the management of convergence insufficiency only. There is insufficient evidence to recommend in-office vision therapy for the management of other types of strabismus, amblyopia, or learning disability in the pediatric population.

## 1. What Is Vision Therapy?

Vision therapy, also known as behavioral optometry, began as a branch of optometry started by Arthur Marten Skeffington to treat visual disorders using methods outside of traditional optometric or orthoptic teaching [1]. As of October 2022, the website for the College of Optometrists in Vision Development offers vision therapy treatments for conditions such as 3D vision problems, convergence insufficiency or excess, crossed eyes, dyslexia, double vision, and amblyopia [2]. The purpose of this paper is to (1) provide a basic overview of the goals of vision therapy, (2) review the evidence behind vision therapy, and (3) offer practical suggestions for pediatricians with parents inquiring about vision therapy.

A typical vision therapy program involves weeks to months of “individually prescribed” exercises done in the office with supplemental exercises at home. They involve various methods to force the use of both eyes, as well as various exercises to improve visual processing, visual tracking, and visual perception. Most of these treatments are non-specific, and some practitioners of vision therapy believe that these visual disorders underly many learning disabilities in children.

While some vision therapy techniques [3,4] have strong scientific evidence to back their use, most treatments do not have strong evidence, and a few may even be harmful to children. One example of a potentially harmful vision therapy is the use of nasal occlusion filters (a form of opaque tape), prescribed by optometrists in the treatment of esotropia (crossed eyes). This tape is placed on the medial aspect of spectacles worn by a child and is known as binasal occlusion. The concept of this technique arises from the debunked “squint masks” [5,6] used to prevent children from using their nasal fields. The authors have encountered this treatment on numerous occasions in children under the age of 5, with parents frequently seeking a second opinion. We have seen multiple children lose valuable time (in some cases, up to a year lost) in the treatment of their strabismus. This time cannot be made up, as there are critical phases in visual development [7]. These children may lose both visual acuity and binocularity, known as stereopsis. 

The problem with binasal occlusion is that children who are born with strabismus, or develop it at an early age, suppress the deviating eye so as not to experience diplopia. Essentially, the brain turns off the deviated eye so that the child can function without double vision. Binasal occlusion serves no purpose to an eye that is already suppressed, and in the fixating eye, the child will just avoid that direction of gaze. It does not correct esotropia, and there are no clinical trials or evidence in literature to support its use in correcting any form of strabismus. A young child has no incentive to straighten a constantly deviating eye because it is constantly suppressed. Surgical intervention, when warranted, at least gives the patient a chance of developing binocularity by straightening the eyes, eliminating suppression.

Some practitioners of vision therapy make use of yoked prisms for the treatment of various neurologic conditions. The concept behind a yoked prism is that the image is moved to a different location than one perceived by the patient, which forces them to look in other areas of their visual fields. This practice is currently used in adults with hemi-spatial neglect or homonymous defects, as well as in children with nystagmus worse in certain directions of gaze. For example, work in adults by Padula et al., demonstrates that the restoration of visual midline using yoked prisms can help with gait [8]. While some studies have demonstrated success in changing posture with yoked prisms, there are others that have demonstrated no significant change in posture in patients using base-down yoked prisms [9]. The use of yoked prisms has not been well studied in children.

There are several other techniques that do not have strong scientific evidence that are used by practitioners engaging in vision therapy. They include techniques such as anti-suppression exercises, sometimes involving filtered lenses, to engage the patients to use their suppressed eye. Anti-suppression has fallen out of favor and can lead to patients experiencing intractable diplopia in their previously suppressed eye. Additionally, anti-suppression does not necessarily change an eye deviation, and the authors have seen surgical intervention in these patients also result in intractable diplopia after straightening an eye that was used to its deviated position. Syntonic therapy involves the applications of certain visible light frequencies into the eyes, which is thought to preferentially treat amblyopia and issues with depth perception. Balance boards involve having patients on a wobbly board that can move forward, backward, or side-to-side in order to improve coordination while performing tasks. Eye tracking or saccade exercises use either electronic targets or computer programs that involve looking at targets and following them in order to work on specific muscles of the eyes. Various colored filters are used to remove certain wavelengths of light, which is theorized to create more eye comfort. The Marsden ball involves setting a ball in motion with letters or numbers in order to involve patients in using their eyes to follow the ball and performs tasks such as calling out the letter or number. Rotation trainers involve the use of a spinning disc with pre-cut holes that patients are asked to do tasks (such as pushing objects into the holes) on while it is spinning. 

## 2. What Is the Evidence for Vision Therapy?

There remains considerable controversy in the use of vision therapy for most pediatric eye disorders. The use of vision therapy to aid in learning disabilities is not supported by the American Academy of Pediatrics, American Academy of Pediatric Ophthalmology and Strabismus, American Academy of Ophthalmology, and American Association of Certified Orthoptists [10,11,12]. Due to the lack of significant evidence demonstrating its efficacy, vulnerable families are required to pay out of pocket, as these frequent office visits are not covered by insurance.

A review of the literature demonstrates a dearth of evidence for the use of vision therapy in children (Table 1). A systemic PubMed search for vision therapy and randomized clinical trial only returned five trials with randomization. All of the other articles were either retrospective review or case series, which makes it difficult to draw appropriate conclusions.

There exists good evidence for certain forms of vision therapy in children [3] and adults [13] with convergence insufficiency. Convergence insufficiency occurs due to an inability or weakness of the eyes to pull in to maintain binocular vision during near work. This is characterized by eye strain, blurred vision, and even headaches after a period of near work. The Convergence Insufficiency Treatment Group [3] randomized pediatric patients with convergence insufficiency into four groups: (1) home-based pencil push-ups, (2) home-based pencil push-ups with vergence and accommodative therapy, (3) office-based vergence and accommodation therapy with home reinforcement, and (4) office-based placebo therapy. They found that patients grouped to office-based vergence and accommodation therapy had lower convergence insufficiency symptoms and better fusional vergence at near distances. However, it is important to note that the in-office treatment group had an additional 60 minutes of treatment time per week [14]. Furthermore, the total cost of the in-office treatment estimated by the authors was $1125.00, which does not account for time off work or school for the parents and students. However, prism glasses, which are sometimes used to help with convergence insufficiency, have been found to be no more effective than reading glasses for treating convergence insufficiency [15].

At this time, there is insufficient evidence to suggest that amblyopia treatment would benefit from in-office-based vision therapy. The only randomized controlled trial completed did not involve in-office therapy, but a home-based video game to improve vision in patients with anisometropic amblyopia (amblyopia due to a significant difference in refractive error between eyes) [16,17]. Pediatric Eye Disease Investigator Group (PEDIG) investigators have attempted to run a randomized clinical trial of vision therapy to treat amblyopia. Study enrolment was limited because in-office vision therapy for amblyopia requires stereopsis and fusion to be performed. Younger children, who are more likely to benefit from amblyopia therapy, have difficulty understanding vision therapy procedures [18]. The danger here is that children who are at a critical stage of visual development may lose valuable time to these unproven therapies and will not be as amenable to treatment as they age. Older children, such as those over the age of 7, who would better understand therapy, unfortunately have a much lower threshold for visual improvement with amblyopia treatment [19]. 

There is no evidence to suggest that vision therapy can aid children who are underperforming at school, such as those with dyslexia, attention deficit disorder, behavioral problems, etc. [20]. Vision problems are not the primary issue in these various learning disabilities and neurocognitive conditions [10]. These patients represent an especially vulnerable group, given the significant stress and burden of these conditions on the family. 

The remainder of the vision therapy literature is formed from uncontrolled, non-randomized, and poorly controlled studies. There are reports of vision therapy being potentially effective in cases of intermittent exotropia. Ma et al., found that vision therapy did not change the amount of misalignment, but improved the control at both far and near distances [21]. However, due to a lack of quality controls, it is difficult to draw conclusions to suggest in-office vision therapy as a treatment. 

There are significant conclusions in the optometric literature to argue against the use of vision therapy in learning disabilities or impaired neurologic function, which is in agreement with the official stance of the many professional organizations that care for children [22,23]. Rucker et al., actively recommended against the use of vision therapy for patients with ADHD or other learning disabilities [24], even in the setting of an increased incidence of convergence insufficiency in ADHD patients [25]. While there are studies that suggest some use in post-concussion patients [26], there were significant confounders given the lack of controls. 

## 3. What Should You Say to Patients Interested in Vision Therapy?

All patients seeking vision therapy and undergoing vision therapy should be evaluated for a second opinion by a pediatric ophthalmologist or an ophthalmologist with experience in caring for children. There is only sufficient evidence to suggest the use of vision therapy in patients suffering from convergence insufficiency, although the data regarding the need for in-office visits has significant confounders. Families of children with impaired vision or learning disabilities should not be encouraged to seek out-of-pocket and potentially unnecessary treatments without clear evidence of benefit. The American Academy of Pediatrics, American Academy of Pediatric Ophthalmology and Strabismus, American Academy of Ophthalmology, and American Association of Certified Orthoptists recommend against the use vision therapy for the treatment of learning disabilities [10,11,12]. For children who do undergo vision therapy, the authors strongly recommend co-treatment or monitoring with a pediatric ophthalmologist, or to seek out a pediatric ophthalmologist if there is no improvement within a few months of therapy in order to prevent further loss of visual development.

## Figures and Tables

**Table 1 children-09-01873-t001:** Summary of all randomized control trials evaluating the effectiveness of vision therapy in children. * The study published in 2005 is a subset of the study from 2008.

Year	Patients	Age (min, max)	Type	Follow-up	Summary	Article
2005	*N* = 72	11.5 ± 2.3	Convergence insufficiency	6 weeks	Base in prism not more effective than reading glasses	Scheinman et al. [13]
2005	*N* = 47	11.5 ± 2.2 (9,18)	Convergence insufficiency	3 months	Office-based vergence therapy outperformed placebo or home-based treatments	Scheinman et al. for the Convergence Insufficiency Treatment Trial Group * [4]
2008	*N* = 221	11.8 ± 2.3 (9, 17)	Convergence insufficiency	3 months	Office-based vergence therapy outperformed placebo or home-based treatments	Convergence Insufficiency Treatment Trial Group * [3]
2013	*N* = 13	N/A	Anisometropic or strabismic amblyopia	N/A	Insufficient patient enrollment	Lyon et al. [14]
2018	*N* = 68	9.9 ± 2.2 (6, 14)	Anisometropic amblyopia	3 months	Monocular video game plus patching enhances visual recovery compared to patching alone	Singh et al. [15]

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
