# Peer review of "Vision Therapy: A Primer and Caution for Pediatricians"

_children, 2022, doi:10.3390/children9121873_

Round 1

Reviewer 1 Report

As a clinical ophthalmologist who has been in practice for 45 years, I can only agree with the authors' conclusions. The publication of this work will certainly help other ophthalmologists in their orientation to the treatment of paediatric patients.

Author Response

Thank you for your review and comments! We truly agree this will help many children.

Reviewer 2 Report

Thank you for allowing me to review the manuscript entitled “Vision Therapy: A Primer and Caution for Pediatricians”.

The manuscript presents a brief review of vision or behavioral therapy and underscores the lack of evidence behind its recommendations. When searching pubmed, there is no trial or study to research the topic, however on the website, like the one below, there are endless claims of the benefits and conditions that can be “improved”, but no reference to a study to prove the claims.

https://www.optometrists.org/vision-therapy/guide-to-vision-therapy/what-is-vision-therapy/

The idea of this manuscript is good; however, I encourage the authors to actually include the resources and the official positions of the scientific community.

https://www.aao.org/eye-health/tips-prevention/vision-training-not-proven-to-make-vision-sharper

https://www.aao.org/clinical-statement/joint-statement-learning-disabilities-dyslexia-vis

https://aapos.org/glossary/vision-therapy

Author Response

We would like to thank our experienced reviewer for excellent comments on our paper. We have updated the manuscript per their comment as below:

1. We have highlighted the following statement on the official statements of AAP, AAPOS and AAO on the evidence for vision therapy. We agree with the reviewer that excessive claims have been made on the benefits of vision therapy.

"The use of vision therapy to aid in learning disabilities is not supported by the American Academy of Pediatrics, American Academy of Pediatric Ophthalmology and Strabismus, American Academy of Ophthalmology, and American Association of Certified Orthoptists.[9]–[11]"
Was updated under what is the evidence for vision therapy.

“The American Academy of Pediatrics, American Academy of Pediatric Ophthalmology and Strabismus, American Academy of Ophthalmology, and American Association of Certified Orthoptists recommends against the use vision therapy for treatment of learning disabilities.[10]–[12]”
Was added to the conclusion to emphasize this point.

Thank you for your consideration,

Edward Kuwera
